# The Current State-Of-Art of Copper Removal from Wastewater: A Review

Nur Hafizah Ab Hamid [1,*], Muhamad Iqbal Hakim bin Mohd Tahir [1], Amreen Chowdhury [1],
Abu Hassan Nordin [1], Anas Abdulqader Alshaikh [1], Muhammad Azwan Suid [1], Nurul 'Izzah Nazaruddin [1],
Nurul Danisyah Nozaizeli [1], Shubham Sharma [2,3] and Ahmad Ilyas Rushdan [1,4,*]

1   School of Chemical and Energy Engineering, Faculty of Engineering, Universiti Teknologi Malaysia,
    Johor Bahru 81310, Johor, Malaysia
2   Mechanical Engineering Department, University Center for Research & Development, Chandigarh University,
    Mohali 140413, Punjab, India
3   Department of Mechanical Engineering, IK Gujral Punjab Technical University, Main Campus-Kapurthala,
    Kapurthala 144603, India
4   Centre for Advanced Composite Materials (CACM), Universiti Teknologi Malaysia (UTM),
    Johor Bahru 81310, Johor, Malaysia
*   Correspondence: nurhafizah.abhamid@utm.my (N.H.A.H.); ahmadilyas@utm.my (A.I.R.)

**Abstract:** Copper is one of the chemical elements that is widely used in various sectors nowadays together with the development of civilization especially in agricultural and industrial sectors. Copper is also considered as one of the heavy metals that is commonly present in wastewater. This preliminary study conducted is mainly focused on the techniques of removal of copper in wastewater. There are a variety of approaches for treating industrial effluent contaminated with heavy metals such as copper. Copper separation can be accomplished using a variety of technologies, each of which has advantages that vary depending on the application. Chemical removal techniques that are commonly used for copper removal are adsorption, cementation, membrane filtration, electrochemical method, and photocatalysis. This study compares the fundamentals and performances of the treatment techniques in addition to the future perspective of copper removal in detail. The study highlights the present research in terms of its strengths and shortcomings, pointing out deficiencies that need to be addressed in future studies, pointing to future research prospects.

**Keywords:** copper removal; adsorption; cementation; membrane filtration; electrochemical; photocatalysis





## 1. Introduction

Copper (Cu) is one of the most important elements and is considered as one of the most widely used metals in various industrial and agricultural practices [1,2]. It is also one of the earliest metals that was collected and utilized and has contributed to an important role in society's survival and improvement since the early days of civilization. Nowadays, copper is currently used in the structure's construction, electricity generation and transmission, electronic product manufacturing, industrial machinery manufacturing, and transportation vehicle manufacturing [3]. However, heavy metals are non-biodegradable, poisonous, and easy to accumulate in living organisms in general, and in humans, at low quantities. It can cause significant disorders such as cancer in the body, harm the neurological system and cause kidney failures and can be fatal due to the high concentration.

Cu is frequently discovered in high concentrations in wastewater as it is widely utilized in the industrial applications, including metal polishing, electroplating, plastics, and etching. Furthermore, even at low concentrations, copper is a highly hazardous metal, and copper-contaminated wastewater must be treated before being discharged into the environment. Heavy metal contamination of soil is becoming a global problem due to the detrimental effects on environmental safety [4]. As Cu becomes more widely used in daily-life purposes, it has a severe influence on environmental degradation and biodiversity in

both aquatic and terrestrial ecosystems, which has unconsciously given an impact toward the human food sources. As a result of global industrialization, soil contamination with harmful metals has increased dramatically in the recent years [5]. Therefore, it is important to use as easy and as effective as possible approaches to treat heavy metals in terms of treatment and removal [6]. Due to their widespread usage in agricultural activities, Cu-based agrochemical products such as fertilizers, pesticides, insecticides, herbicides, fungicides, miticides, and nematicides, which are used to improve crop yield and control plant pests, are usually major sources of Cu deposition in soils [7]. Thus, it is critical to remove the presence of Cu, particularly in water resources, prior to its discharge to the environment [8]. Adsorption, cementation, membrane filtration, electrodialysis, and photocatalysis are the available technologies that have been developed over the years for the removal of copper ions from industrial wastewater.

The major goal of this review paper is to evaluate the fundamentals and key techniques of Cu removal of different types of technologies that are currently applied. Although there are few equivalent review studies on the removal of pollutants, the available literature is broad in scope and takes into consideration other metal ions or is only focused on a single treatment approach. In this study, Cu removal strategies are evaluated in terms of removal efficiency, practicality, environmental friendliness, and process economics.

## 2. Copper (Cu) as a Pollutant in Wastewater

Cu is a heavy metal that can be found abundantly in the earth's crust. Pure copper is very malleable, has a relatively high melting point, and is a good heat conductor [9]. It reacts with oxygen to generate a mild green color when exposed to the environment. Copper alloyed with tin and zinc are also used to make bronze and brass. Since 9000 before century, copper and its alloys have been employed by a variety of civilizations [10]. Copper artifacts have been discovered all over the world. Electrical wiring, plumbing materials, roofing, cookware, vehicle brake pads, and agricultural items are all examples of modern uses for copper. Copper is found in the environment as a result of both natural and human activities. Copper mining, metal and electrical production, agricultural and domestic pesticide and fungicide use, leather processing, and car brake pads are all anthropogenic sources of copper in the environment while forest fires, volcanic eruptions, and windblown dust are the examples of natural sources of copper contamination. Copper is a vital nutrient, but excessive amounts have been linked to stomach and intestinal irritation, liver and kidney damage, and anemia [11]. Wilson's disease patients may be at a higher risk of copper poisoning than the general public [12]. As Cu(II) is a common heavy metal pollutant in water supplies, it may cause hemolysis, cirrhosis, chronic anemia, as well as hepatotoxic and nephrotoxic consequences such as vomiting, cramps, convulsions, and even death. Apart from that, copper has acute effects on aquatic species in marine waters at 4.8 g/L [13]. Cu is particularly toxic to salmon, with deleterious effects occurring at the concentrations of 0.18 to 2.1 g/L in freshwater. It has been reported that the changes in salmonid smoltification processes, interference with fish sensory systems, predator avoidance, juvenile growth, and migratory success behaviors have all been linked to copper [14].

As Cu is non-biodegradable, poisonous, and easy to accumulate in living organisms in general and especially in the human body at low concentrations; they can cause major illnesses such as cancer, nervous system damage, and kidney failure, and can be fatal at high doses [1]. If ingested, heavy metal-polluted water is harmful to one's health and can even be fatal. Cu concentration in drinking water is influenced by pH, hardness, and distribution system availability. Cu contamination of water networks has been reported from both natural and artificial causes. Natural sources of copper in water bodies include weathering of copper-bearing rocks, whereas anthropogenic sources include discharge from urban and industrial wastes, as well as corrosion of pipelines and agricultural application [6]. Cu is a necessary element for living beings, but its presence above the guidelines in drinking water supplies possesses negative health consequences. As a result, to ensure the presence of Cu in water is within the guidelines, Cu removal from water has been accomplished using a

variety of techniques, including ion exchange, precipitation, membrane separation, and adsorption [15–18].

### 3. Copper Removal Techniques

Copper can be removed from wastewater by various techniques, namely, adsorption using natural and modified adsorbents, cementation, membrane filtration, electrochemical methods, and photocatalysis. These techniques work in different ways, but all methods serve the same purpose which is to remove copper from the wastewater. However, different techniques give different results due to many reasons including the efficiency of the method and the parameters that need to be considered when using it. Currently, there is no single technique that can provide completely appropriate treatment, owing to the complicated character of the effluents [19]. Thus, a mix of techniques is frequently utilized to obtain the desired water quality in the most cost-effective manner.

### 3.1. Adsorption

Adsorption technique is the process of mass transfer, where adsorbate, the ions from the gaseous or liquid stream is transferred to the surface of a porous solid, adsorbent as shown in Figure 1. This process creates a film of the adsorbate on the surface of the adsorbent and is bounded by chemical and physical interactions. According to Velasco-Garduño et al., adsorption is currently the most simple and viable method of removing copper from wastewaters [20]. The key reason why adsorption is seen as an adaptable approach for removing copper is due to being easily used on a large scale and has high efficiency. There are also several parameters that influence the adsorption process including surface area, adsorbate initial concentration, solution pH, temperature and interfering chemicals [21].

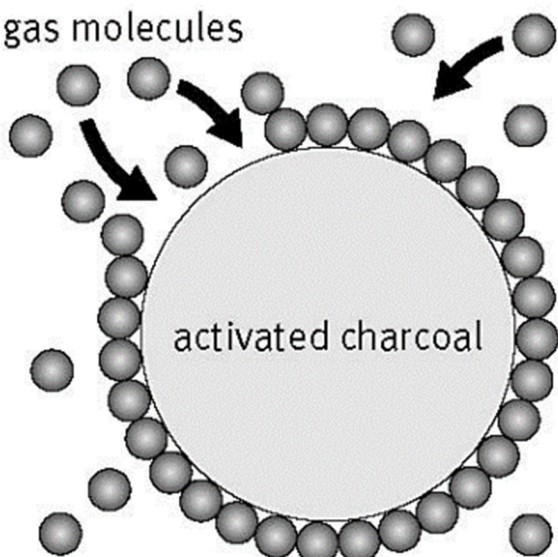

**Figure 1.** Illustration of adsorption of adsorbate onto adsorbent.

However, the development of an effective adsorbent is critical for the technique's effectiveness. Various adsorbents have been obtained from natural materials, modified biopolymers, bio sorbents and nanomaterials [1,22]. Adsorbent materials must have a large internal volume that is accessible to the fluid components being extracted. Next, the adsorbent must have strength and resistance to attrition, as well as excellent kinetic effects, which means it must be capable of swiftly transporting adsorbing molecules to the adsorption sites. The sorption capacity of a material is also determined by its chemical structure, porosity, specific surface area and diffusion characteristics. Furthermore, the adsorption process is regulated not only by adsorbent properties, but also by its shape, chemical composition, water solubility, and pollutant polarity. Last but not least, adsorbents

also need to have numerous functional groups which can bind heavy metal ions from wastewater during removal processes [19,23].

### 3.1.1. Natural Materials as Adsorbents

Natural zeolites with exceptional efficient ion exchange capability are considered the most natural materials for the adsorption of copper [1]. Natural zeolites are porous hydrated aluminosilicate minerals having useful physicochemical features including cation exchange, molecular sieving, catalysis, and sorption [24]. There are over 40 different sorts of natural zeolite found in nature, with clinoptilolite being one of the most common. Natural zeolites with high clinoptilolite content are often used in technological application as it has a high removal capacity of Cu ions. At pH 5, the clinoptilolite performed the best in terms of $Cu^{2+}$ ions removal [25]. However, different types of zeolites have selective adsorption of various copper ions according to Al-Saydeh et al., For example, NaA zeolite removes Cu (III) at natural pH while A4 zeolite is used to adsorb Cu (II) at natural and alkaline pH and the adsorption of Cu (VI) is achieved at acidic pH using the same zeolite. Al-Saydeh et al., also mentioned that clinoptilolite has features of high selectivity for copper ions [1].

Other than that, clay–polymer composites that are made of natural clay minerals can also be used to take out copper ions. Clay minerals are widely used as adsorbent materials deu to their large specific surface area, chemical and mechanical stability, and a wide range of surface and structural features, high cation exchange capacity and low cost [26]. They may be reinforced with a polymeric material to increase their capacity to separate Cu ions from aqueous solutions. Abd Hamid et al. also mentioned that due to the negatively charged membrane surface, the presence of clay particles in the membrane played an important role during adsorption as it can attract positive charge ions, which improved metal adsorption [27]. Natural clay minerals are a good option as the adsorbents for water purification since they are readily available, low-cost, natural resources that are not toxic to the ecosystem. Thus, they will not negatively affect the wastewater when being used for the treatment process.

### 3.1.2. Modified Biopolymers as Adsorbents

Modified biopolymers adsorbents are appealing to industries because they offer several advantages over synthetic polymers, such as great abundance, biodegradability and low cost. In addition, biopolymer-based adsorbents could be regenerated and reused for metal removal after a number of adsorption-desorption cycles [28]. The main advantage of using biopolymers is that its structure contains a variety of functional groups, including amines and hydroxyls, which can improve the effectiveness of heavy metal ion separation [1]. Examples of modified biopolymers that can be employed as adsorbents are cellulose, chitin, chitosan, starch, and lignin.

Cellulose is the primary component of woody plant cell walls. It is the most abundant and widely accessible carbohydrate on the planet. Cellulose has a limited chelating ability to bind heavy metals. However, its modified forms, have extraordinary metal-binding capacities. From the study, they also stated that modification of cellulose can be accomplished either by a direct chemical modification process or by grafting monomer to cellulose backbone. On the other hand, chitin and chitosan are natural amino polysaccharides by which could bind the heavy metals. Physical or chemical modification could be done on chitin and chitosan to enhance the adsorption capacity for heavy metals and their resilience towards the extreme pH conditions [28]. In addition, the modifications could also enhance the mechanical robustness and chemical barrier.

### 3.1.3. Low-Cost Bio Sorbents as Adsorbents

In the recent years, progressive research has been conducted for the removal of heavy metals from industrial wastewater by utilizing the adsorbents produced from agricultural wastes, commonly referred to as bio-sorbents [1]. According to Lu and Gibb, spent grain is one of the low-cost bio sorbents that can be utilized for the adsorption technique [29]. It

is proven to be a safe and biodegradable bio-sorbent for application in the restoration of copper-contaminated wastewater. From the study, they also stated that to avoid competitive inhibition by protons or desorption of $Cu^{2+}$ ions via ion exchange, the pH of the starting solution must be kept above pH 3.6. The total quantity of $Cu^{2+}$ ions adsorbed by wasted grain was generally unaffected by flow rate but increased with $Cu^{2+}$ ion concentration. Aside from that, sawdust could be used as a functional adsorbent for the separation of copper from wastewater as it has a high retention capacity after chemical activation [30].

Apart from that, coconut waste can also be used as a low-cost bio-sorbent. Because of its high tannin content, coconut has a high sorption capability. Next, orange peel can also be utilized to adsorb copper from wastewater. Orange peel is mostly made up of components such as cellulose, pectin and lignin that can help to bind the copper ion and separated them from the wastewater. Adedamola et al., also stated that these components have numerous functional groups, such as carboxyl and hydroxyl, which make them a promising adsorbent material for extracting metal ions from wastewater [31].

On the other hand, biodegradable composite can be one of the alternatives to low-cost bio-sorbents. Velasco-Garduño et al., reported that the Ch-based biodegradable bio-composite can efficiently remove $Cu^{2+}$ in contaminated wastewater allowing for re-use cycles besides having high robustness [20]. The study also shows that Ch-based bio-composites are superior to other commercial absorbents, which are costly, have limited biodegradability, and pose toxicity problems. The use of low-cost adsorbents may aid in the preservation of the surrounding environment.

### 3.1.4. Nanomaterials as Adsorbents

Materials such as carbonaceous nanofibers (CNFs) and graphene oxide (GO) have been applied for the removal of heavy metals from wastewater. Their distinct features, which include strong resistance, good electrical conductivity, and thermal stability, as well as their huge specific surface areas, make them viable adsorbents. The results reported by García-Díaz et al., prove to demonstrate that helical carbon nanofibers are a viable copper adsorbent in wastewaters. The pH value has the biggest impact on copper adsorption capacity among the factors which were tested at a pH of 10. For GO, Al-Saydeh et al. stated that the maximum copper ion sorption capacity was found to be 45.2 mg/g on pure GO [1,32].

Moving on, nanocomposites made from clay minerals and polymers have developed as a revolutionary method for cleaning polluted water and are employed as a coagulant, catalytic agent, membrane, or filter. The addition of clay in appropriate amounts with numerous inorganic polymers and biopolymers in the nanoscale range, or nano clay, for the development of nanocomposites boosted its efficacy for pollutant removal in aqueous systems [32]. However, to improve the environmental sustainability and safety of created nanocomposites, the use of cost-effective and environmentally friendly resources such as agro-waste, green extract, and industrial by-products should be encouraged.

### 3.2. Cementation

Cementation is a simple method for removing and recovering dangerous or valuable metals in their metallic and reusable condition. Cementation involves the electrochemical precipitation of one metal by another metal that is more electropositive. In the case of copper removal, cementation is the process in which the copper ions are reduced to zero valances at the interface of iron by spontaneous electrochemical reduction to reach the copper metallic state [1]. The cementation is a cost-effective process as it is simple to operate. However, an excessive sacrificial metal consumption is the drawback of this cementation approach [33]. Furthermore, the sacrificial metal dissolution, which is more critical at low pH levels is also the disadvantage of this technique. Thus, one critical feature of this technique is the use of a non-toxic metal or a metal that is already present in the solution, in order to avoid contamination with another ion [34].

This technique's efficiency is influenced by many factors including the pH value, amount of sacrificial metal, copper concentration, temperature, stirring speed, the presence of other ions, size of zinc particles and type of atmosphere. Based on Panão et al., the cementation with zinc is favorable to be used to remove copper from solutions containing zinc and iron compared to the use of iron and aluminium. This is because zinc is a superior precipitant as it has a larger oxidation potential than iron. Despite having a larger oxidation potential than zinc, the protective oxidation layer generated on the surface of aluminium resulted in worse outcomes [34].

### 3.3. Membrane Filtration

Membrane filtration has gained lots of attention in the recent years for its ability to remove suspended particles, organic molecules, and inorganic substances such as heavy metals from copper-contaminated industrial waste water [8]. Furthermore, the key benefits of using a membrane technique are no phase change, economic productivity, simplicity of scaling up, high separation efficiency, and environmental safety. Ultrafiltration (UF), nanofiltration (NF), and reverse osmosis (RO) are the commonly used membrane filtration technique to remove copper from the wastewater.

#### 3.3.1. Ultrafiltration (UF)

At low transmembrane pressures, ultrafiltration (UF) is a viable membrane technology for removing copper ions from sewage. The pore size of UF is 5–20 nm, and the molecular weight (MW) of isolated particles should be in the 1000–100,000 Da range. Based on the membrane's properties, UF membranes may achieve a good removal rate of over 90% for initial copper concentrations ranging from 10 to 160 mg/L, with the optimal pH range being 5.5 to 6, and an operating pressure of 2 to 5 bars [35]. The high packing density of UF is a significant benefit, as it reduces the amount of space required.

Polymer enhanced ultrafiltration (PEUF) has also been studied in the recent years [35]. PEUF has been suggested as a method for removing copper ions from industrial effluent, as shown in Table 1. The incorporated polymers are responsible for binding metal ions and forming macromolecular metallic complexes that the UF can reject. These polymers are normally polyelectrolytes containing functional groups, such as polyethyleneimine (PEI), carboxyl methyl cellulose (CMC), and poly (acylic acid) sodium, which may interact with ions by electrostatic attraction or establish stable bonds with them. The metal to polymer ratio, polymer type, and pH levels are just a few of the many variables that influence the PEUF process. Progressive research has been conducted to find the best polymer to be used in the PEUF approach. PEI, humic acid, polyacrylic acid (PAA), and diethylaminoethyl cellulose were among the polymers examined by Fu et al., that exhibited selective separation of copper ions with higher energy efficiency. Molinari et al., demonstrated that polyethyleneimine (PEI) is successful in removing Cu (II) from polluted wastewater, with the greatest results achieved when the pH was greater than 6 and the metal to polymer mass ratio was about 3. PEUF has a plethora of benefit, including strong affinity selectivity and efficacy in the elimination process. PEUF has yet to find widespread industrial use, despite a substantial amount of research and several articles in this sector.

**Table 1.** Cu (II) removal by PEUF membrane.

| UF Type | Membrane Type | Surfactant Agent | Initial Conc. | Ideal pH | Removal Efficiency | Reference |
|---------|---------------|------------------|---------------|----------|--------------------|-----------|
| PEUF | Polyethersulfone | PEI | 50 mg/L | pH > 6 | 94% | [36] |
| PEUF | Polyethersulfone | Carboxy methyl cellulose | 10 mg/L | pH = 7 | 97.6% | [37] |
| PEUF | Ceramic | Poly (acylic acid) sodium | 160 mg/L | pH = 5.5 | 98–99.5% | [38] |

### 3.3.2. Nanofiltration

The use of nanofiltration (NF) for the removal of heavy metals has grown quickly in recent years, owing to the fact that it solves some of the issues associated with traditional removal methods [39]. Al-Rashdi et al., proved the effectiveness of the NF membrane in the removal of heavy metal ions. The NF is a pressure-driven process with particle diameters ranging from ultrafiltration (UF) to reverse osmosis (RO). Donnan exclusion (charge repulsion) and size exclusion are two distinct separation processes found in the NF membrane [40]. The NF membrane has a number of advantages over other membrane types, including a better rejection of multivalent copper ions than the UF membrane [24]. NF membranes also have a greater water permeability, a higher refusal, and a reduced pressure than RO membranes. As a result, the NF membrane approach is widely regarded as a low-energy way of eliminating heavy metal ions. NF membranes generally have a particle size of 1 nm, which corresponds to a MW cut-off (MWCO) of 300–500 Da, according to Mohammad et al., Many different membrane separation technologies have been developed to minimize the upfront cost and solving the issues of toxic elements in the industrial water [41]. According to Kotrappanavar et al., most NF membranes are made up of thin layer fiber composites of several manufactured materials with electrostatic interactions. These groups have the potential to improve the polymer's ability to remove charged heavy metal ions from effluent. Electro migration, as well as sieving, the Donnan effect, solution diffusion, and dielectric exclusion, are employed in NF membranes to separate electrically neutral and charged particles, making them ideal for removal. According to Al-Rashdi et al., under pressures ranging from 3 to 5 bar and pH values ranging from 1.50 to 5, the NF membrane completely recovered copper ions in a 1000 mg/L copper solution, suggesting the feasibility of NF membranes for copper ions refusal. When the concentration of copper rises to 2000 mg/L, however, the capacity of NF membranes to reject copper ions decreases. Table 2 compares the removal of Cu (II) utilizing various NF technologies under various operating circumstances. It can be observed that the integrated NF/RO membranes have the capability to remove high concentrations of Cu (II) under low pressure. These combination membranes can be used to treat wastewater with high rejection and pure water recovery.

**Table 2.** Cu (II) removal techniques.

| Membrane Types | Initial Concentration | Removal Efficiency | Operation Circumstance | References |
|---|---|---|---|---|
| NF | 0.01 M | 47–66% | Transmembrane pressure 1–3 bar | [42] |
| NF | 0.47 M | 96–98% | Pressure = 20 bars | [43] |
| RO | $7.86 \times 10^{-3}$ M | 98–99.5% | Pressure = 5 bars | [44] |
| RO | Between $4.7 \times 10^{-4}$ and $1.57 \times 10^{-3}$ M | 70–90% | Low pressure RO | [45] |
| RO+NF | 2 M | More than 95% | Pressure = 35 bars | [46] |
| RO+NF | 0.015 M | 95–99% | Pressure = 3.8 bars | [47] |

### 3.3.3. Reverse Osmosis

Reverse osmosis (RO) is a method of separating polluted wastewater by pushing it through a membrane that rejects the impurities solely on a single side while allowing the clean solution to flow through on the other (Figure 2) [35]. Over the last decade, reverse osmosis (RO) has emerged as one of the most effective separation technologies for industrial effluent treatment and groundwater recharge [44]. The majority of the separation occurs in the polymer matrix of RO membranes, which has a substantial protective film. On a broad scale, the RO process may be used to eliminate a range of compounds from dirty water, especially in industrial settings. However, Gunatilake shows that the RO process is mostly composed of diffusive processes, and that the separation efficiency is highly dependent on water flow rate, pressure, and solute concentration. RO brine is typically rich in heavy metals (such as Mo, Cu, and Ni) as well as other less harmful toxic metals (i.e., Zn and Fe) [44]. Hybrid systems combining electrodialysis and a particulate reactor are the best alternative for increasing RO recovery by treating its core since the pellet reactor reduces scalability capability while the electrodialysis treats the dirty wastewater [44]. Research has

been conducted on the efficacy of the RO method for copper removal and determined that extraction efficiencies of 70–99.9% may be achieved. In a lab-scale experiment which integrate the bioreactor system with RO membranes, Dialynas and Diamadopoulos observed that the elimination of copper ions was successfully removed.

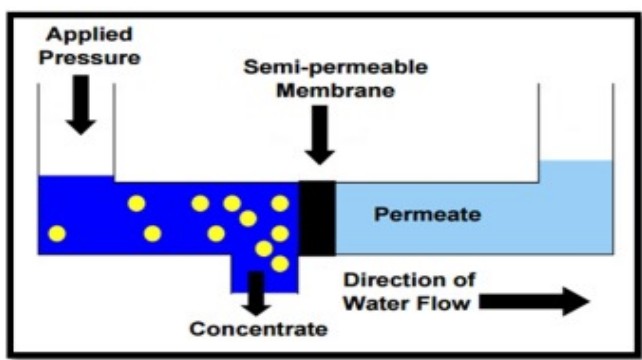

**Figure 2.** Reverse osmosis mechanism [35].

*3.4. Electrochemical Methods*

In metallurgical and metal processing, electrochemical removal techniques for separating metal ions have been commonly adopted [48]. Electrocoagulation and electrodialysis are the two most common electrochemical procedures for removing heavy metal ions. Electrodialysis is an electrochemical process to separate copper ions from industrial effluent, by which the copper ions exchange is done via surfaces under an electric field [49]. Furthermore, the electric field starts when surface activities generate hydroxyl ions at the cathode and protons at the anode, respectively. Metal ions have a significant attraction to be desorbed and moved more towards the cathode by electromigration because the ionic transformation of protons is much bigger than that of hydroxyl ions [50]. Ur Rahman et al., invented a cost-effective and efficient electrolytic–electrodialytic equipment and procedure for recovering metal ions from municipal wastewater. According to Nasef et al., the features of the membrane used in the electrodialysis mechanism must be examined since all these factors impact the extent of copper ions segregation. Caprarescu et al., also looked at the requirements for the membrane utilized in the electrodialysis process, which included chemical and thermal stability, as well as the capability to perform the separation process at high temperatures and in solutions with extremely high or low pH values. A constant speed of metal-contaminated effluent may be recovered in a single unified cell flow battery, as shown in Figure 3. There are three major portions in each individual cell: the catholyte section (101), the metal-contaminated wastewater section (102), and the anolyte section (103). In order to access all cells from (108) and exit from the metal-contaminated wastewater stream (100) will be split into many streams (109). The anolyte portion will receive the metal ions via (110), while the oxygen generated by the anode and the hydrogen produced by the cathode both exit from (112). (113). The stream will contain all effluent wastewater (114).

Electrocoagulation, according to Lakhsamanan et al., is a great method for heavy metals removal, particularly Cu, from industrial effluent. One of the electrochemical methods for removing copper ions from industrial effluent is electrocoagulation [40]. This is depending on dissolved metal anode from Al, Fe, or hybrid Al/Fe electrode materials in situ to produce coagulant [51]. Metal ions are generated at the anode, while hydrogen gas is formed at the cathode, which can assist in floating the agglomerates particles out of the water. Electrocoagulation provides a number of advantages over alternative treatments, including the fact that it is quick, efficient, economical, and ecologically friendly [52]. The available density is the main factor in the electrocoagulation process since it impacts several variables such as bubble generation rate, coagulant dosing rate, and floc size or growth, in which these factors might affect electrocoagulation effectiveness. When the present density rises, the anode dissociation rate rises as well, resulting in a rise in the quantity of

copper hydroxide flocs and a rise in copper percentage removal [53]. Table 3 demonstrates the impact of the electrocoagulation process operating conditions, electrode composition, current or existing density, ideal pH, solution conductivity, and energy usage on copper removal rate.

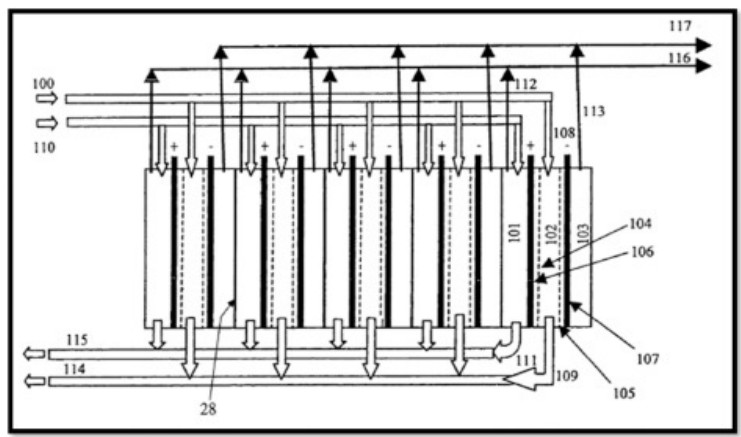

**Figure 3.** The principle of electrodialysis cell used in continuous process [1].

**Table 3.** Different electrocoagulation techniques for removing Cu(II).

| Reactor | Current Density | Conductivity (mS/cm) | Expenditure of Energy | Ideal pH | Electrode Component | Removal Rate | References |
|---|---|---|---|---|---|---|---|
| Continuous | 4.8 A/dm3 | - | - | 4 | Al-Al | 99% | [54] |
| | 5 A | - | 10.99 kWh/kg | 0.64 | Ss-Ti | 98.8% | [55] |
| | 5 A | 1600 | 35.63 kWh/g | 7 | Fe-Fe | 99.99% | [56] |
| | 5 A | 1600 | 35.06 kWh | 7 | Al-Al | 99.9% | [56] |
| Batch | 0.3 A | 0.634 | - | 5 | RO-Ti-Ss | 99% | [57] |
| | 100 A/m$^2$ | 2 | 10.07 kWh/m$^3$ | 3 | Fe-Al | 100% | [28] |
| | 33 A/m$^2$ | 20 | - | 9 | Al-Al | >50% | [58] |

Despite the significant numbers of study and method advances in electrocoagulation technology over the last decade, more research is required to fathom the implications of cell design and electrode configuration on heavy metal removal efficiency. Moreover, because the majority of recent research has been executed on a laboratory level, extra work has to be done into assessing the electrocoagulation mechanism at the production plant scale.

*3.5. Photocatalysis*

Photocatalysis is a promising treatment technology for a variety of municipal wastewaters. Under solar radiation, electron–hole pairs (e/h+) can be continuously created from semiconductors in this process. ZnO, ZnS, TiO$_2$, and CeO$_2$ are just a few of the semiconductors that have been used [59]. In order to discover the optimum semiconductor for removal of copper, numerous experiments have been reported in various studies. According to Mahdavi, et al. [60], TiO$_2$ is the most extensively used because of its strong photocatalysis, durability, low toxicity, and exceptional dielectric characteristics. A photocatalytic procedure to remove Cu (II) has been reported utilizing TiO$_2$ as a semiconductor and a 254 nm UVC light in varied test conditions. The best results for Cu (II) removal were obtained when the pH was between pH 3.5 and pH 4.5 and the TiO$_2$ mass was between 0.5 and 0.75 g. TiO$_2$ can be employed in the photocatalysis removal process roughly 80% of Cu (II) according to the findings [1]. Furthermore, Barakat et al., performed a photocatalytic degradation experiment utilizing UV-irradiated TiO$_2$ suspension to remove copper and destroy complex cyanide. Copper with an initial concentration of 102 M was fully eliminated in 3 h, according to the data. Wahyuni et al., investigated the photocatalytic elimination of Cu (II) using an UV light with a spectrum of 290–390 nm and a starting concentration of 10 mg/L. When 50 mg of TiO$_2$ was used at pH 5, the greatest photocatalytic efficiency was achieved, removing 45.56% of the copper ions.

### 3.6. Comparison of Copper Removal Processes

In general, alternatives for removing copper ions from industrial wastewater have undergone progressive research. Figure 4 summarizes the process of copper removal from wastewater using different techniques that have been discussed previously. Chemical, physical, and biological treatments are the most common, and they are chosen for a variety of reasons, including high selectivity, ease of control, and reduced space requirements. The most acceptable treatment methods for removing copper ions from industrial effluents are physico-chemical treatments. However, due to the high cost of the chemicals required and the significant energy consumption, they still have a high operational cost. Since metal-contaminated effluent can contain both inorganic and organic contaminants, photocatalysis is a promising technology for eliminating them at the same time. Therefore, every method of treatment does have its own benefits and drawbacks. The key benefits and drawbacks of the various physicochemical procedures considered in this analysis are represented in Table 4, with their Cu(II) removal efficiencies shown in Table 5.

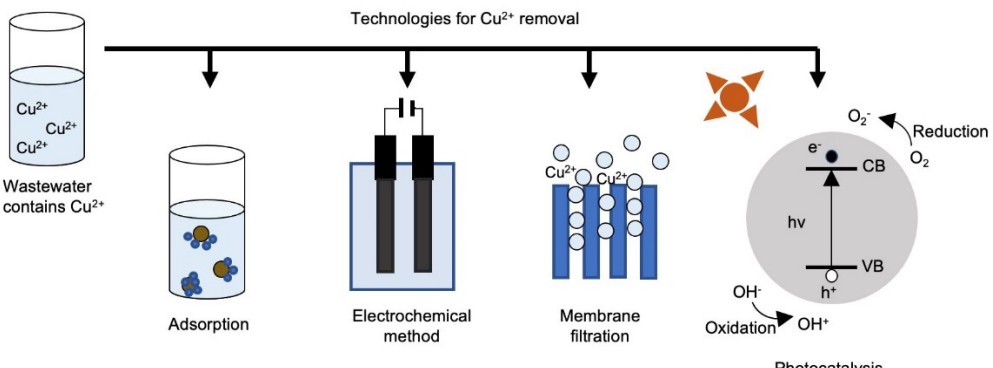

**Figure 4.** The process of copper removal from wastewater.

**Table 4.** Advantages and disadvantages of different treatment methods.

| Removal Technique | Benefits | Drawbacks |
|---|---|---|
| Adsorption using inexpensive adsorbents | • Minimal start-up costs<br>• Simple concept | • Copper ions are restricted to certain concentrations. |
| Cementation | • Inexpensive cost procedure<br>• Low energy usage<br>• Simple functioning and convenient | • Excessive use of sacrificial metal |
| Membrane filtration | • Low operating pressure<br>• Small space requirement<br>• Highly selective separation | • Operating costs are high. |
| Electrochemical methods | • Separation that is extremely selective<br>• Environmentally safe | • A substantial operating cost<br>• Extremely high energy usage |
| Photocatalysis | • Removal of chemical and metal contaminants in tandem<br>• Creates less harmful by-products | • The time span is rather long.<br>• It is only beneficial in situations. |

**Table 5.** Cu (II) removal efficiency using different treatment methods.

| Removal Technique | Type of Material | Operating Condition | Removal Efficiency | Ref. |
|---|---|---|---|---|
| Adsorption using inexpensive adsorbents | • Orange peel<br>• Activated carbon derived from Ascophyllum nodosum seaweed | Initial concentration of 100 mg/L, adsorbent dosage of 1 g, reaction time of 120 min, pH 5.<br>Initial concentration of 500 mg/L, adsorbent dosage of 0.2 mg, reaction time of 6 hr, pH 5. | 86.3%<br><br>99.0% | [61]<br><br>[62] |

**Table 5.** *Cont.*

| Removal Technique | Type of Material | Operating Condition | Removal Efficiency | Ref. |
|---|---|---|---|---|
| Cementation | • Iron Powder | Initial concentration of 750 mg/L, reaction time of 7 min, pH 2.95, linear speed of 24.13 m/s. | 95% | [63] |
| | • Zinc powder | Initial concentration of 3 g/L, reaction time of 10 min, pH 1, rotational speed of 500 rpm. | 90% | [64] |
| Membrane filtration | • PEUF+polyvinylamine (PVA) | Initial concentration of 25 mg/L, Pressure of 200 kPa, pH 6 | 90% | [65] |
| | • NF: Poly (amidoamine) dendrimer (PAMAM) | Initial concentration of 1000 mg/L, Pressure of 1000 kPa, pH 8 | 99% | [66] |
| Electrochemical methods | • Nickel Sulfate | Initial concentration of 0.06 M, pH 6.6, energy supply 10 V, 20 h | 70% | [67] |
| | • Graphene oxide/polypyrrole | Initial concentration of 100 mg/L, pH 7.0, energy supply 0.6 V, 96 h | 82.8% | [68] |
| Photocatalysis | • TiO2/ZnO–CaAlg | Initial concentration of 20 mg/L, reaction time of 2 h, pH 7 | 98.9% | [69] |
| | | Initial concentration of $10^{-4}$ mol/L, reaction time of 30 min, pH 5 | 70% | [70] |

## 4. Challenges of Copper Removal

As discussed before, copper (Cu) is removed from wastewater by different methods for examples chemical, physical, and biological. However, the selection is influenced by numbers of factors with the consideration of advantages and disadvantages. For the current methods applied, the challenges are copper concentration limits, high operating costs, excess consumption of sacrificial metals, high energy consumption, and the long time required for the process [1]. In addition, in the presence of other metals, the selectivity of copper could be a problem, by which most copper is bound with other metals in wastewater. For example, the work done by Larsson and his coworkers—on real acid mine drainage solutions with copper and other metals in acidic conditions using PEI-GA-DE to extract copper—showed that the selectivity was low, resulting in a high copper concentration after the process [71]. In addition, Abdel Salam et al., reported a limited applicability to particular copper ion concentrations where some polluted samples are not treated due to the low concentration of copper which could be the case. A study by Anastopoulos and Kyzas reported that the efficiency of heavy metals removal via the adsorption method is also dependent on the pH of the treated water, in which the optimum pH was reported to be in the range of pH 2 to pH 6. As for cementation technology, DemirkIRan and Künkül mentioned the fundamental problem is the excessive use of sacrificial metal. The excess usage of a source always causes unsustainability which is considered a drawback and a challenge to solve in order to improve the current solution in removing copper from wastewater [1].

Some difficulties arise as a result of the type of reactor used, especially when using electrochemical technology. Using a fluidized bed, for example, could result in poor charge transfer and non-uniform potential distribution. In addition, the chemical-physical treatment introduces new challenges such as incomplete removal, consumption of high energy and converting the dissolved metals to solid sludge which requires further waste treatment [72].

## 5. Conclusions

In conclusion, due to the negative effects of heavy metals on human health and living creatures in the environment, the removal of heavy metals in general and copper from industrial effluent is a very essential aspect of most contemporary environmental research. Different treatment strategies such as physical, chemical, and biological treatments were discussed based on the past few years. These methods that have been introduced are adsorption, cementation, membrane filtration, electrodialysis, and photocatalysis. However, there are some precautions that should be taken seriously as it is influenced by the parameters such as initial concentration of copper ions, pH values, economic parameters such as operation cost, and the environmental effects and compatibility of each of the various

methods conducted. More studies should be conducted in order to improve on certain areas. This is because current cementation agents require a long time to remove greater copper ions from wastewater; hence, future research in these areas should focus on evaluating new cementation agents that can shorten the process time. Furthermore, the impact of pressure as a control parameter has yet to be determined. Innovative strategies are required to produce economical, readily available, excellent, and long-lasting membranes for membrane filtering. In the case of electrodialysis, innovative designs are required to improve separation efficiency. More research into the effect of temperature on the photocatalysis approach is required. It is necessary to develop catalysts for a high-photon-efficiency process that can use a wider range of spectrum. Recently, studies have demonstrated that biosorption and natural adsorbents can be applied to treat copper pollutants from wastewaters. Consequently, new forms of biosorbents must be evaluated to achieve maximum effectiveness. Some of the benefits include low cost of raw material, good adsorption performance, and environmentally friendly. Therefore, it is a promising technique for copper removal. For further investigation into copper treatment in wastewater, various information gaps must be investigated. First off, laboratory batch equilibrium studies have been the main focus of the majority of copper biosorption research. This is due to two factors: (a) the lack of bulk biomass for use at full scale, and (b) batch procedures, which are simple to conduct in a lab setting but challenging in the field. Therefore, there is clearly a need for further research into the development of low-cost biosorbents with excellent adsorption capacities for copper treatment. Additionally, it seems that the pH impact is a limiting factor that influences copper treatment possibilities. For instance, the starting solution's pH must be maintained above pH 3.6 to prevent competitive inhibition by protons or the desorption of $Cu^{2+}$ ions by ion exchange. Undesirably, pH adjustment may be expensive, thus it should be carefully taken into account when designing the copper treatment procedure. Despite having a significant potential to efficiently treat copper in wastewater, other procedures including cementation, membrane filtration, electrodialysis, and photocatalysis may not be fully implemented due to the expense.

**Author Contributions:** Conceptualization, N.H.A.H. and M.I.H.b.M.T.; validation, A.C., A.H.N., A.A.A. and A.I.R.; writing—original draft preparation, N.H.A.H., M.I.H.b.M.T. and A.C.; writing—review and editing, A.H.N., A.A.A., M.A.S., N.'I.N., N.D.N., S.S. and A.I.R.; funding acquisition, N.H.A.H. and A.I.R. All authors have read and agreed to the published version of the manuscript.

**Funding:** The authors would like to express gratitude for the financial support received from the Universiti Teknologi Malaysia, the project "The impact of Malaysian bamboos' chemical and fibre characteristics on their pulp and paper properties, grant number PY/2022/02318—Q.J130000.3851.21H99". The research has been conducted under the program Research Excellence Consortium (JPT (BPKI) 1000/016/018/25 (57)) provided by the Ministry of Higher Education Malaysia (MOHE).

**Data Availability Statement:** Not applicable.

**Conflicts of Interest:** The authors declare no conflict of interest.

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
