# Peer review of "The Current State-Of-Art of Copper Removal from Wastewater: A Review"

_water, doi:10.3390/w14193086_

Round 1
Reviewer 1 Report
This preliminary study compares the fundamentals and performances of the treatment
techniques, in addition to the future perspective of copper removal in detail, focusing on the techniques of removal of copper considered as one of the heavy metals that are commonly present in wastewater. Copper removal from wastewater is made using a variety of technologies, each of which has advantages that vary depending on the application. These are mainly adsorption, cementation, membrane filtration, electrochemical method, and photocatalysis. The study evaluates these methods in terms of their strengths and deficiencies that could be useful in future studies.
This is a valuable paper and could be published after minor remarks as below.
General Remarks
The Conclusions are overall. Some detailed data comparing discussed methods of Cu removal should be added.
Abstract. Most important data comparing discussed methods of Cu removal should be added.
Detailed remarks
1. Point 2, paragraph 1 – some references should be added
2. Figure 1 – clarify please the title
3. Figure 3 should be self-explaining. Please add proper information.
4. Tables 1 and 2 should be self-explaining. Please add proper information.
Author Response
Thanks for your letter and the thoughtful comments from the referees about our paper entitled “The Current State-of-Art of Copper Removal from Wastewater: A Review”. We carefully analysed all the comments and these comments are very valuable and helpful for perfecting and modifying our manuscript, and also have important guiding significance for our research. Therefore, we carefully checked the manuscript and revised it according to each comment. Consequently, we feel that our manuscript is substantially strengthened. The detailed corrections in the paper and the responses to the reviewer’s comments are as the following list of revisions.

Reviewer 2 Report
The work deals with the overviewing the most recent methods applied for the copper removal from the environmental effluents namely wastewater.
Yet has been studied in myriad of works, copper is still being an element-of-interest for many operators as it is one of the most important environmental compound in the nature.
Please, follow the down below remarks:
1. Some of important citations are missing in the introduction:
“Copper (Cu) is one of the most important elements and is considered as one of the most widely used metals in various industrial and agricultural practices [1, https://doi.org/10.1016/j.talanta.2012.05.063].“
“As a result of global industrialization, soil contamination with harmful metals has increased dramatically in the recent years [4]. Therefore, it is important to use as easy and as effective as possible approaches to treat heavy metals in terms of treatment and removal [https://doi.org/10.1039/C2AY25133G]. Due to their widespread usage in agricultural activities, Cu-based agrochemical products such as fertilizers, pesticides, insecticides, herbicides, fungicides, miticides, and nematicides, which are used to improve crop yield and control plant pests, are usually major sources of Cu deposition in soils [5].”
2. Please, make a comparison table, including the most recent methods for the wastewater removal to confront their removal efficacy.
3. Please, make a figure of general workflow of copper removal to provide the reader quick glance on how it looks. It should involve main steps of removal process.
4. Please, add to conclusion some of future aims of the authors in this scope of research. I see general future points are already provided.
Author Response

(The authors gave the same response as above.)

Round 2
Reviewer 2 Report
Authors have reacted to the given remarks. Thus, I consider publication at current state.